# Three New Species of *Gongronella* (*Cunninghamellaceae*, *Mucorales*) from Soil in Hainan, China Based on Morphology and Molecular Phylogeny

**DOI:** 10.3390/jof9121182

**Published:** 2023-12-10

**Authors:** Yi-Xin Wang, Heng Zhao, Zi-Ying Ding, Xin-Yu Ji, Zhao-Xue Zhang, Shi Wang, Xiu-Guo Zhang, Xiao-Yong Liu

**Affiliations:** 1College of Life Sciences, Shandong Normal University, Jinan 250358, China; wyx13953348060@163.com (Y.-X.W.); 15270343451@163.com (Z.-Y.D.); ji15965902393@163.com (X.-Y.J.); wangssdau@126.com (S.W.); zhxg@sdau.edu.cn (X.-G.Z.); 2State Key Laboratory of Mycology, Institute of Microbiology, Chinese Academy of Sciences, Beijing 100101, China; zhaoheng21@bjfu.edu.cn; 3School of Ecology and Nature Conservation, Beijing Forestry University, Beijing 100081, China; 4Shandong Provincial Key Laboratory for Biology of Vegetable Diseases and Insect Pests, College of Plant Protection, Shandong Agricultural University, Taian 271018, China; zhangzhaoxue2022@126.com

**Keywords:** soil-born fungi, *Mucoromycota*, taxonomy, new taxa, molecular phylogeny

## Abstract

The genus *Gongronella* is important in agriculture and industry by secreting various natural bioactive metabolites such as chitosanases and organic acids. During the most recent 8 years, a total of 14 new species have been described, remarkably enriching the diversity of this genus. In this study, we added three more new species to this valuable genus, based on a combination of morphological traits and phylogenetic information. Six strains of the genus *Gongronella* were isolated from soil collected in Hainan Province, China. Phylogenetic analyses of ITS and LSU rDNA sequences grouped these strains into three independent clades. According to their unique morphological characteristics, they were classified as *G. multiramosa* sp. nov., *G. qichaensis* sp. nov. and *G. oleae* sp. nov. The *G. multiramosa* was characterized by multiple branched sporangiophores and was closely related to *G. pedratalhadensis*. The *G. qichaensis* was characterized by obscure collars and closely related to *G. butleri*, *G. hydei* and *G. banzhaoae*. The *G. oleae* was characterized by the presence of oil droplets in the sporangiospores and was closely related to *G. chlamydospora* and *G. multispora*. Their descriptions and illustrations were provided, and their differences from morphological allies and phylogenetic-related species are discussed.

## 1. Introduction

Fungi in the genus *Gongronella* Ribaldi have an important application by producing a variety of bioactive secondary metabolites. *Gongronella butleri* (Lendn.) Peyronel & Dal Vesco is well-known in the chitosan industry [1,2]. Two chitosanases Csn1 and Csn2 were purified from *Gongronella* sp. JG [3,4]. The Csn2 was identified as a new enzyme, a potential candidate for the preparation of oligosaccharides from colloidal chitosans [3]. *Gongronella* spp. were found to possess phosphate-solubilizing abilities [5] and to degrade metalaxyl [6]. *Gongronella* sp. strain W5 penetrated the root cells of Chinese kiwi, improving its phosphate acquisition and consequently promoting its growth by secreting organic acids [7]. As non-laccase-producing fungi, *Gongronella* spp. can induce *Panus rudis* Epicr to overproduce laccase, maximizing its laccase-producing capacity [8]. A *Gongronella* species was also found to counter its competitor *Coprinopsis cinerea* (Schaef.) Redhead, Vilgalys & Moncalvo by secreting antifungal metabolites [9].

*Gongronella* Ribaldi belongs to the phylum *Mucoromycota* Doweld, class *Mucoromycetes* Doweld, order *Mucorales* Dumort and family *Cunninghamellaceae* Naumov ex R.K. Benj. [10]. It was established in 1952 to accommodate the type species only, *G. urceolifera* Ribaldi [11]. Three years later, the second species was proposed in this genus, namely *G. butleri* (Lendn.) Peyronel & Dal Vesco, which was recombined from *Absidia butleri* Lendn. Another seven years passed, and the third species was reported as *G. lacrispora* Hesselt. & J.J. Ellis. Since then, no more species in this group of fungi have been described for half a century. Traditionally, the main diagnostic criteria for the genus *Gongronella* have been the presence of distinct swollen apophyses, globose sporangia and constricted columellae [12]. Currently, a consensus has been reached that fungi are classified on the basis of a combination of multiple molecular phylogenetic analyses and morphological features [13,14,15]. Consequently, the new species of *Gongronella* have been increased. These newly found species include the following: *G. guangdongensis* F. Liu, T.T. Liu & L. Cai 2015; *G. koreana* Hyang B. Lee & T.T.T. Nguyen 2015; *G. orasabula* Hyang B. Lee, K. Voigt, P.M. Kirk & T.T.T. Nguyen 2016; *G. brasiliensis* C.A.F. de Souza, D.X. Lima & A.L. Santiago 2017; *G. sichuanensis* Z.Y. Zhang, Y.F. Han, W.H. Chen & Z.Q. Liang 2019; *G. zunyiensis* C.B. Dong, Y.F. Han & Z.Q. Liang 2019; *G. eborensis* M.R. Martins, C. Santos, C. Soares, Cl. Santos & N. Lima 2020; *G. hydei Doilom* 2020, *G. namwonensis* Hyang B. Lee, A.L. Santiago & H.J. Lim 2020; *G. pedratalhadensis* L.W.S. Freitas, H.B. Lee & A.L. Santiago 2020; *G. banzhaoae* Y.P. Tan & Bishop-Hurley 2023; *G. chlamydospora* H. Zhao, Y.C. Dai, Yuan Yuan & X.Y. Liu 2023; *G. multispora* H. Zhao, Y.C. Dai, Yuan Yuan & X.Y. Liu 2023; and *G. pamphilae* Y.P. Tan, Bishop-Hurley & R.G. Shivas 2023. The genus *Gongronella* currently consists of 16 species, with the type species *G. urceolifera* being treated as a synonym of *G. butleri* [16].

Six strains of *Gongronella* were isolated from soils in Hainan Province, China. According to rDNA molecular phylogenetic analysis and morphological comparison, these strains were classified into three new species.

## 2. Materials and Methods

### 2.1. Isolation and Morphological Observation

In 2023, soil samples were collected in Hainan Province, and pure strains were isolated from these samples by combining soil dilution plates and single spore isolation methods. Exactly 10 g of soil samples were transferred into a conical bottle containing 90 mL sterile water and mixed with a shaker at 120 rpm for 20 min to prepare soil suspension. One milliliter of the suspension was pipetted into 9 mL sterile water to obtain 10^−2^ soil suspension. The previous step was repeated to obtain 10^−3^ and 10^−4^ soil suspensions. A 200 μL of 10^−3^ and 10^−4^ soil suspensions were pipetted onto the center of Rose Bengal Chloramphenicol agar (RBC: peptone 5.00 g/L, Glucose 10.00 g/L, KH_2_PO_4_ 1.00 g/L, MgSO_4_·7H_2_O 0.50 g/L, Rose Bengal 0.05 g/L, chloramphenicol 0.10 g/L, agar 15.00 g/L) [17], dispersed evenly with a sterilized triangle glass spatula and cultivated at 25 °C in the dark for 2–5 days. Subsequently, the agar with fungal mycelia at the edge of the colony was transferred to a new Potato Dextrose Agar (PDA: glucose 20.00 g, potato 200.00 g, agar 20.00 g, sterilized water 1000.00 mL and pH7) and macroscopically photographed on the 7th day with a digital camera (Canon PowerShot G7X, Canon, Tokyo, Japan).

Microscopic morphological characteristics of fungi were observed with a stereoscope (Olympus SZX10, OLYMPUS, Tokyo, Japan) and a light microscope (Olympus BX53, OLYMPUS, Tokyo, Japan), and photographed with a high-definition color digital camera (Olympus DP80 OLYMPU, Tokyo, Japan). All strains were stored in 10% sterilized glycerin at 4 °C. Morphologies were statistically calculated from 30 measurements per character [18]. Living cultures (including ex-types) were deposited in the China General Microbiological Culture Collection Center, Beijing, China (CGMCC) and the Shandong Agricultural University Culture Collection, Taian, China (SAUCC). Dried type specimens were deposited in the Herbarium Mycologicum Academiae Sinicae, Beijing, China (HMAS). Taxonomic information for the new taxa has been registered in the Fungal Name repository (https://nmdc.cn/fungalnames/, accessed on 25 October 2023).

### 2.2. DNA Extraction and Amplification

Genomic DNA was extracted from mycelia using the CTAB (cetyl trimethyl ammonium bromide) method. Additionally, a rapid extraction plant DNA magnetic beads kit was employed (GeneOn BioTech, Germany; http://www.geneonbio.com/content/?1153.html/, accessed on 24 October 2023) [19,20]. The internal transcribed spacer (ITS) of rDNA was amplified with primers ITS5 (5’- GGA AGT AAA AGT CGT AAC AAG G -3’)/ITS4 (5’- TCC TCC GCT TAT TGA TAT GC -3’) [21]. LSU rDNA was amplified with primers LR0R (5’- GTA CCC GCT GAA CTT AAG C -3’)/LR7 (5’- TAC TAC CAC CAA GAT CT -3’) [22]. Polymerase chain reaction (PCR) procedures included a pre-denaturation at 95 °C for 5 min, and then 35 cycles of denaturation at 95 °C for 50 s, annealing at 47 °C for 30 s and an extension at 72 °C for 1.5 min, and, finally, an extra extension at 72 °C for 10 min. Amplification was performed in a final volume of 20 μL reaction mixture, containing 10 μL 2 × Hieff Canace^®^ Plus PCR Master Mix (Yeasen Biotechnology, Shanghai, China, Cat No. 10154ES03), 0.5 μL of forward and reverse primers each (10 μM) (TsingKe, Qingdao, China), 1 μL of template genomic DNA (about 1 μM), and 8 μL of distilled deionized water. The PCR products were checked by electrophoresis with 1% agarose gel and stained with GelRed. The fragments were visualized under ultraviolet light at 254 nm [18]. A gel extraction kit (Cat. AE0101-C; Shandong Sparkiade Biotechnology Co., Ltd., Jinan, China) was then used for gel recovery. Sanger sequencing was carried out by Biosune Company Limited (Shanghai, China). Consensus sequences were assembled using MEGA v.7.0 (Mega Limited, Auckland, New Zealand) [23]. All sequences generated in this study were deposited at GenBank (Bethesda, Rockville Pike, USA) under the accession numbers in Table 1.

### 2.3. Phylogenetic Analyses

The obtained DNA sequences were searched using BLAST with default parameters for closely related items against the NCBI GenBank nucleotide database [24]. Newly generated sequences and their related sequences retrieved from GenBank (Table 1) were aligned using MAFFT 7 online services (http://mafft.cbrc.jp/alignment/server/, accessed on 20 October 2023) [25], employing default parameters for accurate alignment. Each marker was first analyzed individually (ITS or LSU) and then jointly (ITS-LSU), with maximum likelihood (ML) and Bayesian inference (BI) algorithms integrated with the CIPRES science portal (https://www.phylo.org/, accessed on 20 October 2023) [26]. ML was performed with RaxML-HPC2 on XSEDE (8.2.12) [27] and 1000 fast bootstrap repeats were performed using the GTRGAMMA model of nucleotide evolution. For BI, the optimal evolutionary model for each partition was determined using MrModeltest v.2.3 (accessed on 12 May 2022) [28] and included in the analysis. BI was performed with MrBayes on XSEDE (3.2.7a) [29,30,31]. For ML analysis, the default parameters were used and BI was performed using a fast boot algorithm with an automatic stop option. Bayesian analysis consisted of 5,000,000 generations of four parallel runs with stop rule options and a sampling frequency of 100 generations. The burnin score was set to 0.25 and the posterior probability (PP) was determined from the remaining trees. All resulting trees were drawn using FigTree v.1.4.4 (http://tree.bio.ed.ac.uk/software/figtree, accessed on 20 October 2023), and the layout of the trees was carried out with Adobe Illustrator CC 2019 (https://adobe.com/products/illustrator/, accessed on 20 October 2023 ).

## 3. Results

### 3.1. Phylogenetic Analyses

The sequence matrix included 26 strains from 16 species of *Gongronella* with *Cunninghamella echinulata* CBS 156.28 selected as an outgroup for phylogenetic comparison. A total of 1888 characters comprised ITS rDNA (1–909) and LSU rDNA (910–1888). Among them, there were 1207 constant, 426 variable but parsimony non-informative and 255 parsimony informative characters (Appendix A). MrModelTest suggested that the Dirichlet fundamental frequency and GTR+I+G evolution pattern for both partitions were adopted in Bayesian inference. The topology of the Bayesian tree was consistent with that of the ML tree and therefore was used as a representative to summarize the evolutionary history within the genus *Gongronella* (Figure 1). The *G. multiramosa* was closely related to *G. pedratalhadensis* with a well support (MLBV = 100, BIPP = 1.00). The *G. qichaensis* was closely related to *G. butleri* (BIPP = 0.74), *G. hydei* and *G. banzhaoae*. The clade of *G. oleae* was a sister clade to *G. chlamydospora*/*G. multispora*.

### 3.2. Taxonomy

#### 3.2.1. *Gongronella multiramosa* Yi Xin Wang, H. Zhao & X.Y. Liu, sp. nov., Figure 2

Fungal Name—No: FN 571690

**Figure 2 jof-09-01182-f002:**
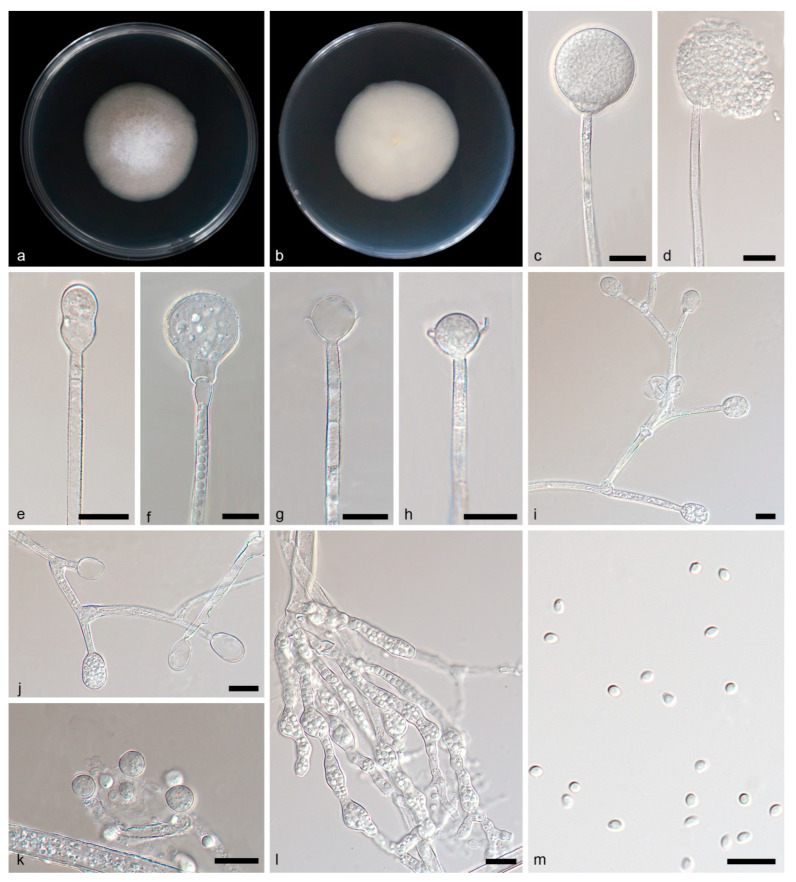
*Gongronella multiramosa* ex-holotype CGMCC 3.26216. (**a**,**b**) Colonies on PDA ((**a**) obverse; (**b**) reverse); (**c**,**d**) unbranched sporangiophores with sporangia; (**e**) a developing sporangium with a septum; (**f**) an immature sporangium and a line of oil droplets in the sporangiophore; (**g**,**h**) columellae, collars and septa; (**i**) a branched sporangiophore with immature sporangia; (**j**) a branched sporangiophore with immature sporangia and sterile (aborted) sporangia; (**k**) giant cells; (**l**) rhizoids; (**m**) sporangiospores; bars: (**c**–**m**) 10 μm.

Etymology—The epithet “*multiramosa*” (Latin) refers to the multiple branched sporangiophores.

Type—China, Hainan Province, Changjiang Li Autonomous County, Bawangling National Forest Park (19°12′ N, 109°84′ E), from a soil sample, 9 April 2023, Yi-Xin Wang (Holotype HMAS 352645, Ex-holotype strain CGMCC 3.26216).

Description—Rhizoids are hyaline, branched, irregularly shaped and filled with oil droplets. Stolons absent. Sporangiophores on aerial mycelia, erect or slightly curved, unbranched, or sympodially branched up to 7 times, 4.68–128.44 × 2.57–3.94 μm, hyaline, smooth, mostly aseptate on lateral branches, usually with a septum near apophyses on main axes, occasionally containing a line of oil droplets. Sterile (aborted) sporangia sometimes on the top of short lateral branches of sporangiophores, ovoid, 9.60 × 6.15 μm in diameter. Fertile sporangia hyaline or light yellow, spherical, 15.49–23.17 μm in diameter, smooth and deliquescent walled, leaving a collar after releasing sporangiospores. Columellae mostly hemispherical and 3.61–5.69 × 8.01–9.81 μm, sometimes sub-hemispherical and 3.02–3.93 × 7.64–9.98 μm, hyaline, smooth. Collars distinct, 1.30–7.23 μm. Apophyses hyaline, smooth, variously shaped, mostly hemispherical and 4.43–5.55 × 8.53–9.02 μm, partially cup shaped and 4.62–6.97 × 8.47–10.03 μm. Sporangiospores are not uniform, hyaline, smooth, sub-spherical and 1.72–2.57 μm in diameter, ovoid and 2.57–3.25 × 1.74–2.29 μm, unusually reniform and 2.66–3.40 × 1.31–1.91 μm. Chlamydospores present, ellipsoidal or fusiform. Giant cells intercalary, globular, sub-spherical, 3.03–6.68 μm in diameter. Zygospores not found.

Culture characteristics—Colonies on PDA in darkness at 25 °C growing slowly, reaching 21.62–25.57 mm in diameter in 7 days, white, regular at the edge and cottony in the center, reversely milky white.

Additional specimen examined—China, Hainan Province, Changjiang Li Autonomous County, Bawangling National Forest Park (19°11′ N, 109°83′ E), from a soil sample, 9 April 2023, Yi-Xin Wang (SAUCC 4056-4).

GenBank accession numbers—CGMCC 3.26216 (OR733546 for ITS rDNA and OR733611 for LSU rDNA); SAUCC 4056-4 (OR733545 for ITS rDNA and OR733610 for LSU rDNA).

Notes—Based on phylogenetic analysis of ITS-LSU rDNA sequences, the two isolates of the new species *Gongronella multiramosa* formed an independent clade with full supports (MLBV = 100, BIPP = 1.00; Figure 1), which is closely related to *G. pedratalhadensis* (MLBV = 100, BIPP = 1.00; Figure 1). However, this new species differs morphologically from *G. pedratalhadensis* in sporangiophore, sporangium, septum, columella, apophysis, chlamydospore and zygospore [32]. The *G. multiramosa* has a longer sporangiophore (4.68–128.44 × 2.57–3.94 μm vs. 9.5–30 × 2.5–7 μm), which is branched more times (up to 7 vs. only 1–2) than *G. pedratalhadensis*. The sporangium of *G. multiramosa* is smaller than that of *G. pedratalhadensis* (15.49–23.17 vs. 17.00–35.00 μm). The *G. multiramosa* has fewer septa on sporangiophores compared to *G. pedratalhadensis* (0–1 vs. >2). Although *G. multiramosa* is similar in shape of columellae to *G. pedratalhadensis*, it is smaller in size (3.02–5.69 × 9.74–9.98 μm vs. 5.00–15.00 × 4.00–21.50 μm). The *G. multiramosa* is remarkably different from *G. pedratalhadensis* in shape and size of apophyses: The former is mostly hemispherical or cup shaped and 4.43–5.55 × 8.53–9.02 μm, and the latter is tube shaped and 5.00–15.00 × 4.50–15.00 μm. Chlamydospores are present in *G. multiramosa* but not in *G. pedratalhadensis*. Combining morphological and molecular phylogenetic analyses, we classified the two isolates together as a new species: *G. multiramosa* allied to *G. pedratalhadensis*.

#### 3.2.2. *Gongronella qichaensis* Yi Xin Wang, H. Zhao & X.Y. Liu, sp. nov., Figure 3

Fungal Name—No: FN 571691

**Figure 3 jof-09-01182-f003:**
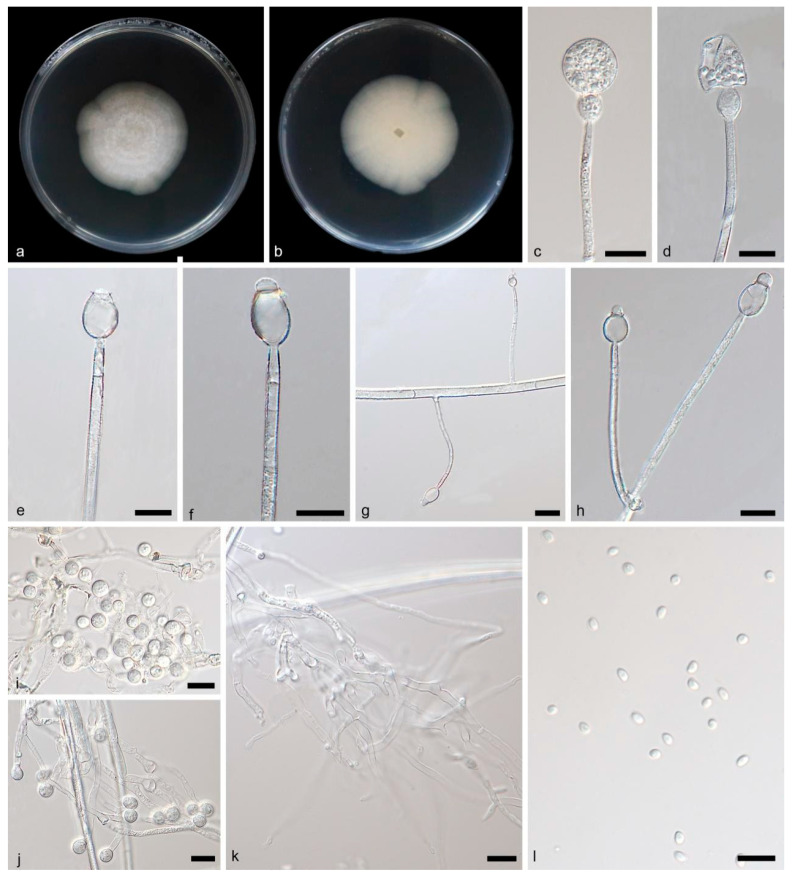
*Gongronella qichaensis* ex-holotype CGMCC 3.26218. (**a**,**b**) Colonies on PDA ((**a**) obverse, (**b**) reverse); (**c**,**d**) unbranched sporangiophores with sporangia; (**e**,**f**) columellae, collars and septa; (**g**) sporangiophores on an aerial hypha, showing septa in both hypha and sporangiophores; (**h**) branched sporangiophores; (**i**) intercalary giant cells; (**j**) terminal giant cells; (**k**) rhizoids; (**l**) sporangiospores; bars: (**c**–**l**) 10 μm.

Etymology—The epithet “*qichaensis*” (Latin) refers to the location where the type was collected, Qicha Town.

Type—China, Hainan Province, Changjiang Li Autonomous County, Qicha Town (19°12′ N, 109°15′ E), from a soil sample, 9 April 2023, Yi-Xin Wang (Holotype HMAS 352646, Ex-holotype strain CGMCC 3.26218).

Description—Rhizoids are hyaline, branched and irregularly shaped. Stolons absent. Sporangiophores on aerial mycelia, erect or slightly curved, unbranched, or branched 1–2 times, 17.33–141.19 × 0.72–4.25 μm, hyaline, smooth, usually aseptate, sometimes with one septum below apophyses, occasionally with two septa. Sterile (aborted) sporangia sometimes on the top of short lateral branches of sporangiophores, ovoid, 12.15–13.68 μm in diameter. Fertile sporangia hyaline or light yellow, spherical, 7.93–36.65 μm in diameter, smooth and deliquescent walled, leaving a slight collar after releasing sporangiospores. Columellae mostly ellipsoidal and 0.79–6.50 × 1.21–8.10 μm, sometimes sub-hemispherical to curved and 0.96–2.04 × 2.53–4.52 μm, hyaline, smooth. Collars distinct, 0.82–1.82 μm. Apophyses hyaline, smooth, variously shaped, mostly pear shaped to oval, 4.55–13.37 × 3.36–10.70 μm, partially elliptical or sub-spherical and 5.97–11.25 × 4.81–9.01 μm. Sporangiospores not uniform, hyaline, smooth, mostly ellipsoidal, 3.03–4.24 × 2.05–2.79 μm, sometimes reniform, 2.76–3.65 × 2.32–2.82 μm, occasionally spherical, 2.43–3.30 μm in diameter. Chlamydospores present, ellipsoidal or fusiform. Giant cells intercalary and terminal, globular, sub-spherical, 3.49–6.68 μm in diameter. Zygospores not found.

Culture characteristics—Colonies on PDA in darkness at 25 °C growing slowly, reaching 20.33–22.69 mm in diameter in 7 days, white, cottony, regular at edge, reversely milky white.

Additional specimen examined—China, Hainan Province, Changjiang Li Autonomous County, Qicha Town (19°12′ N, 109°15′ E), from a soil sample, 9 April 2023, Yi-Xin Wang (SAUCC 4137-3).

GenBank accession numbers—SAUCC 4137-3 (OR733543 for ITS rDNA and OR733606 for LSU rDNA); CGMCC 3.26218 (OR733544 for ITS rDNA and OR733607 for LSU rDNA).

Notes—Based on phylogenetic analyses of ITS-LSU rDNA sequences, the two isolates of the new species *G. qichaensis* formed an independent clade with high supports (MLBV = 99, BIPP = 0.99; Figure 1), which is closely related to *G. butleri* (BIPP = 0.74; Figure 1), *G. hydei* and *G. banzhaoae*. Due to the unavailability of descriptions and illustrations for *G. banzhaoae*, we compare only *G. butleri* and *G. hydei* with the new species G. *qichaensis*. This new species differs morphologically from *G. hydei* in the colony, rhizoid, sporangiophore, sporangium, columella, apophysis and giant cell [5]. The *G. qichaensis* is smaller in colonial diameter than *G. hydei* (20.33–22.69 mm vs. 60.00–65.00 mm). The *G. qichaensis* differs from *G. hydei* in the width of sporangiophores (0.72–4.25 μm vs. 1.60–3.20 μm). The *G. qichaensis* is bigger in sporangium than *G. hydei* (7.93–36.65 μm vs. 10.00–18.8 μm). Although *G. qichaensis* is similar in shape to columellae to *G. hydei*, it is different in size (0.79–6.50 × 1.21–8.10 μm vs. 1.70–4.70 × 2.20–6.30 μm). The *G. qichaensis* is remarkably different from *G. hydei* in shape and size of apophyses: The former is mostly pear shaped to oval or elliptical or sub-spherical, 4.55–13.37 × 3.36–10.70 μm, and the latter is cup shaped or cuboid shaped, 2.50–7.30 × 3.50–7.80 μm. The *G. qichaensis* is smaller in giant cells than *G. hydei* (3.49–6.68 μm vs. <25.00 μm). This new species differs morphologically from *G. butleri* in sporangiophore, apophysis, sporangium, sporangiospore and giant cell [2,11]. The *G. qichaensis* is narrower in sporangiophore than *G. butleri* (0.72–4.25 μm vs. 3.0–5.5 μm). The *G. qichaensis* is remarkably different from *G. butleri* in shape: The former is mostly pear shaped to oval, while the latter is mostly hemispherical or cup shaped. The *G. qichaensis* is different from *G. butleri* in sporangial size (7.93–36.65 μm vs. 11.50–24.40 μm). Although *G. qichaensis* is similar in shape of apophyses to *G. butleri*, it is smaller in size (2.4–4.2 × 2.0–3.3 μm vs. 3.5–7.2 × 6.7–8.5 μm). Giant cells are present in *G. qichaensis* but not in *G. butleri*. Combining morphological and molecular phylogenetic analyses, we classified the two isolates as a new species: *G. qichaensis* allied to *G. hydei*, *G. butleri* and *G. banzhaoae*.

#### 3.2.3. *Gongronella oleae* Yi Xin Wang, H. Zhao & X.Y. Liu, sp. nov., Figure 4

Fungal Name—No: FN 571693

**Figure 4 jof-09-01182-f004:**
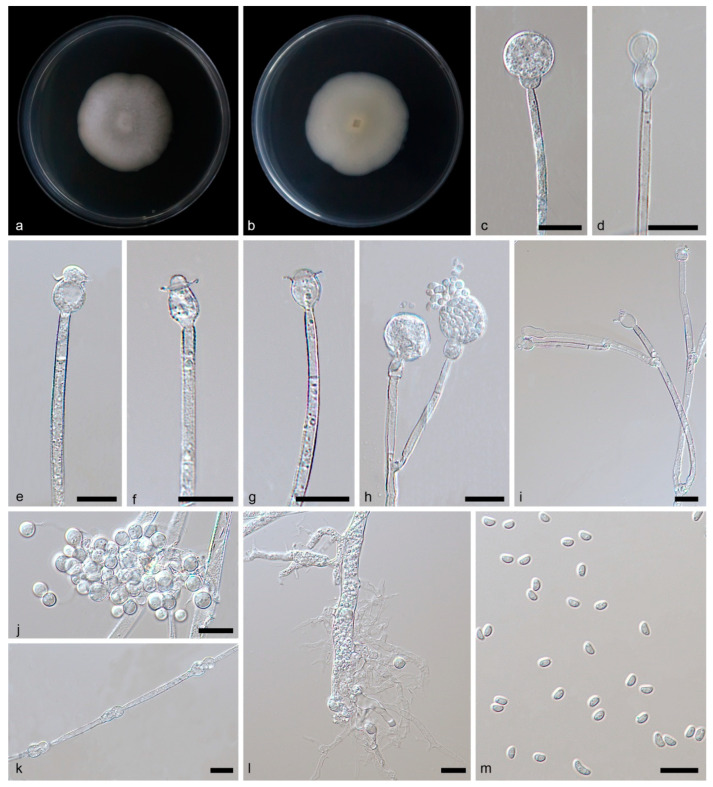
*Gongronella oleae* ex-holotype CGMCC 3.26217. (**a**,**b**) Colonies on PDA ((**a**) obverse; (**b**) reverse); (**c**) an unbranched sporangiophore with a sporangium; (**d**) an aborted sporangium with a septum; (**e**–**g**) columellae, collars and septa; (**h**) branched sporangiophores with sporangia, sporangiospores and septa; (**i**) branched sporangiophores with an aborted sporangium and some fertile sporangia; (**j**) terminal giant cells; (**k**) chlamydospores; (**l**) rhizoids; (**m**) sporangiospores; bars: (**c**–**m**) 10 μm.

Etymology—The epithet “*oleae*” (Latin) refers to the oil droplets in the sporangiospores.

Type—China, Hainan Province, Changjiang Li Autonomous County, Jianfeng Town (18°71′ N, 108°87′ E), from a soil sample, 10 April 2023, Yi Xin Wang (Holotype HMAS 352647, ex-holotype strain CGMCC 3.26217).

Description—Rhizoids are hyaline, branched, irregularly shaped and filled with oil droplets in the main stem. Stolons absent. Sporangiophores on aerial mycelia, erect or slightly curved, unbranched, or branched 3–4 times, 6.99–96.84 × 0.89–3.47 μm, hyaline, smooth, mostly aseptate, sometimes with one septum on branches. Sterile (aborted) sporangia sometimes on the top of short lateral branches of sporangiophores, sub-spherical, 6.95–7.84 μm in diameter. Fertile sporangia hyaline or light yellow, spherical, 8.75–24.54 μm in diameter, smooth and deliquescent walled, leaving a collar after releasing sporangiospores. Columellae are mostly sub-spherical or ovoid and 2.64–5.17 × 3.22–6.47 μm, sometimes hemispherical, and 0.41–3.26 × 2.79–5.34 μm, hyaline, smooth. Collars distinct, 0.73–7.42 μm. Apophyses hyaline, smooth, variously shaped, pear shaped and 4.43–5.55 × 8.53–9.02 μm, cup shaped and 4.62–6.97 × 8.47–10.03 μm, elliptical or sub-spherical, 2.69–8.00 × 2.83–9.07 μm. Sporangiospores not uniform, hyaline, smooth, ovoid, 2.40–3.34 × 1.51–2.35 μm, reniform, 2.58–4.99 × 1.48–2.24 μm, with one or two oil droplets. Chlamydospores cucurbit shaped and 11.23–22.56 × 4.26–9.75 μm in diameter. Giant cells terminal, globular, sub-spherical, 3.21–6.47 μm in diameter. Zygospores not found.

Culture characteristics—Colonies on PDA in darkness at 25 °C growing slowly, reaching 16.30–17.00 mm in diameter in 7 days, white, regular at the edge and cottony in the center, reversely milky white.

Additional specimen examined—China, Hainan Province, Changjiang Li Autonomous County, Jianfeng Town (18°71′ N, 108°87′ E), from a soil sample, 10 April 2023, Yixin Wang (SAUCC 4164-2).

GenBank accession numbers—CGMCC 3.26217 (OR742078 for ITS rDNA and OR733608 for LSU rDNA); SAUCC 4164-2 (OR742079 for ITS rDNA and OR733609 for LSU rDNA).

Notes—Based on phylogenetic analyses of ITS-LSU rDNA sequences, the two isolates of the new species *G. oleae* formed an independent clade with full supports (MLBV = 100, BIPP = 1.00; Figure 1), which is closely related to *G. chlamydospora* and *G. multispora*. However, this new species differs morphologically from *G. chlamydospora* and *G. multispora*. As for *G. chlamydospora*, they are differentiated by growth speed, sporangiophore, sporangium and chlamydospore [33]. The *G. oleae* grows slower than *G. chlamydospora* (2 mm/day vs. 8 mm/day). The *G. oleae* was branched more times (up to 4 vs. 1–2) than *G. chlamydospora*. The *G. oleae* is bigger in sporangium than *G. chlamydospora* (8.75–24.54 μm vs. 8.50–17.00 μm). The *G. oleae* is remarkably different from *G. chlamydospora* in shape and size of chlamydospores. As for *G. multispora*, the differences are shown in growth speed, sporangium, apophysis, columella and chlamydospore [33]. The *G. oleae* grows slower than *G. multispora* (2 mm/day vs. 7 mm/day). The *G. oleae* is bigger in sporangium than *G. multispora* (8.75–24.54 μm vs. 12.00–17.00 μm). Although *G. oleae* is similar in shape of apophyses to *G. multispora*, it is smaller in size (2.69–8.00 × 2.83–9.07 μm vs. 8.00–12.00 × 7.00–9.50 μm). The *G. oleae* varied more in columella shape than *G. multispora*: The former is mostly sub-spherical or ovoid and sometimes hemispherical, while the latter is only hemispherical. Chlamydospores are present in *G. oleae* but not in *G. multispora*. Combining morphological and molecular phylogenetic analyses, we classified the two isolates as a new species *G. oleae* allied to *G. multispora* and *G. chlamydospora*.

## 4. Discussion

Located on the northern edge of the tropics, China’s Hainan Province falls into the Indo-Burma biodiversity hotspots and is known as a natural greenhouse with extraordinary fungal diversity. Based on molecular and morphological data, three new species of *Gongronella* isolated from soil habitats in southern Hainan are introduced herein.

*Gongronella* was initially erected to accommodate the *Absidia*-like fungus, *G. urceolifera* (currently a synonym of *G. butleri*), which is characterized by its spherical protrusions [34]. However, this genus has remained one of the most understudied taxa of fungi for dozens of years. There has been an increase in research activity since 2015 and more and more new species have been described in this genus. Currently, there are 19 species in this genus including the 3 new species proposed herein, all of which were listed in Table 1. All members of this group of fungi were isolated from soil samples (Table 1), and therefore this genus is probably called soil-born fungus. The 26 *Gongronella* strains used in this study are distributed all over the world, including Australia (2 strains), Brazil (5), China (11), Korea (5), Portugal (2) and the UK (1), and, consequently, this genus is most likely a worldwide fungus, which was confirmed by 3341 samples and 25,840 sequence variants filtered from the GlobalFungi database (https://globalfungi.com/, accessed on 6 December 2023; Asia, 72.13%, North America, 12.57%, South America, 6.55%, Europe, 3.32%, Africa, 2.87%, Australia, 2.39% and Pacific Ocean, 0.15%). According to these statistics, the investigation of this genus in China seems to be more detailed than in other regions.

It is worth noting that the ITS and LSU sequences of *Gongronella pamphilae* were not found in NCBI, so the phylogenetic evolutionary tree was constructed in this study without the inclusion of *G. pamphilae* sequences. *Gongronella* taxonomy used to be studied primarily on the basis of morphological features and the phylogeny of ITS rDNA sequences [35,36]. When *G. sichuanensis* was published in 2019, the LSU rDNA sequences were added, resulting in a consistency with previous ITS phylogeny [13]. Since then, taxonomists studying this genus have adopted the practice of combining ITS-LSU rDNA with morphology.

Phylogenetic analysis of the six strains using ITS and LSU rDNA sequences revealed three robust monophyletic clades. Compared with *Gongronella pedratalhadensis*, the new species *G. multiramosa* has longer sporangiophores, more branching frequency, smaller sporangia and larger apophyses. No chlamydospores are found in *G. pedratalhadensis*, but *G. multiramosa* found. The new species *G. qichaensis* forms rhizoids, which are not found in its relatives *G. hydei* and *G. banzhaoae*. The sporangial size of *G. qichaensis* is larger than that of *G. hydei*, and the apophyses of *G. qichaensis* are significantly different from those of *G. hydei* and *G. banzhaoae*. Lastly, *G. qichaensis* lacks obvious collars, while *G. butleri* and *G. hydei* exhibit remarkable collars. The sporangial diameter of *G. oleae* is larger than that of *G. chlamydospora*, while the apophysis width of *G. oleae* is smaller than that of *G. chlamydospora*. Giant cells have also been found in *G. oleae* but not described in *G. chlamydospora*. These notable morphological dissimilarities plus those phylogenetically independent clades ensure their novelty.

In summary, the molecular phylogenetic and morphological results support that the six strains described in this paper represent three new species, *G. multiramosa*, *G. qichaensis* and *G. oleae*.

## Figures and Tables

**Figure 1 jof-09-01182-f001:**
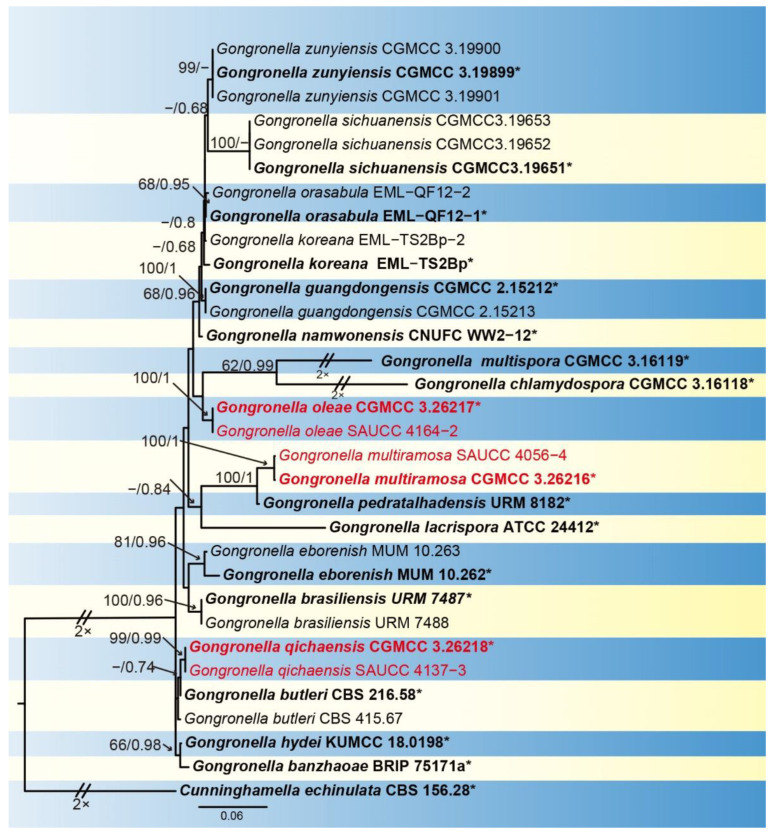
A maximum likelihood (ML) phylogenetic consensus tree inferred from ITS and LSU rDNA sequences, showing relationships among species of *Gongronella* with *Cunninghamella echinulata* CBS 156.28 as outgroup. The maximum likelihood bootstrap value (MLBV) and Bayesian inference posterior probability (BIPP) are successively shown at the nodes and separated by a slash “/”. Strains marked with a star “*” and bolded represented are ex-types or ex-holotypes. The strains isolated and sequenced in this study are shown in red. Branches shortened to fit the page are represented by double slashes “//” and folds “×”. The scale in the bottom center indicates 0.06 substitutions per site.

**Table 1 jof-09-01182-t001:** Information of specimens used in this study.

Species	Strains	Substrates	Countries	GenBank Accession Numbers	
				ITS	LSU
*G. banzhaoae*	BRIP 75171a *	Soil	Australia	OR271908	OR259049
*G. brasiliensis*	URM 7487 *	Soil	Brazil	NR_155148	KY114932
	URM 7488	Soil	Brazil	KY114931	KY114933
*G. butleri*	CBS 415.67	Soil	Brazil	JN206288	MH870714
	CBS 216.58 *	Soil	UK	JN206285	MH869292
*G. chlamydospora*	CGMCC 3.16118	Soil	China	OL678157.1	n.a.
*G. eborensis*	MUM 10.262 *=CCMI 1100 *	Soil	Portugal	KT809408	MN947301
	MUM 10.263=CCMI 1101	Soil	Portugal	GU244500	MN947302
*G. guangdongensis*	CGMCC 2.15212 *	Soil	China	NR_158464	MN947303
	CGMCC 2.15213	Soil	China	KC462740	MN947304
*G. hydei*	KUMCC 18.0198	Soil	China	NR_171964	MT907273
*G. koreana*	EML-TS2Bp *	Soil	Korea	KP636529	KP636530
	EML-TS2Bp-2	Soil	Korea	KP835545	KP835542
*G. lacrispora*	ATCC 24412 *	Soil	Brazil	GU244498	JN206609
** *G. multiramosa* **	**CGMCC 3.26216 ***	**Soil**	**China**	**OR733546**	**OR733611**
	**SAUCC 4056-4**	**Soil**	**China**	**OR733545**	**OR733610**
*G. multispora*	CGMCC 3.16119	Soil	China	OL678158.1	n.a.
*G. namwonensis*	CNUFC WW2-12	Soil	Korea	NR_175640	MN658482
** *G. oleae* **	**CGMCC 3.26217 ***	**Soil**	**China**	**OR742078**	**OR733608**
	**SAUCC 4164-2**	**Soil**	**China**	**OR742079**	**OR733609**
*G. orasabula*	EML-QF12-1 *	Soil	Korea	NR_148087	KT936263
	EML-QF12-2	Soil	Korea	KT936270	KT936264
*G. pamphilae*	BRIP 74936a	Soil	Australia	n.a.	n.a.
** *G. qichaensis* **	**CGMCC 3.26218 ***	**Soil**	**China**	**OR733544**	**OR733607**
	**SAUCC 4137-3**	**Soil**	**China**	**OR733543**	**OR733606**
*G. pedratalhadensis*	URM 8182	Soil	Brazil	MN912512	MN912508
*G. sichuanensis*	CGMCC 3.19651 *	Soil	China	MK813373	MK813855
	CGMCC 3.19652	Soil	China	MK813374	MK813856
	CGMCC 3.19653	Soil	China	MK813375	MK813857
*G. zunyiensis*	CGMCC 3.19899 *	Soil	China	MN453856	MN453853
	CGMCC 3.19900	Soil	China	MN453857	MN453854
	CGMCC 3.19901	Soil	China	MN453858	MN453855
*Cunninghamella echinulata*	CBS 156.28	n.a.	n.a.	JN205895	MH877699

Notes: New species established in this study are in bold. Ex-type or ex-holotype strains are labeled with a star mark “*”. The abbreviation of “n.a.” stands for “not available”.

## Data Availability

The sequences from the present study were submitted to the NCBI database (https://www.ncbi.nlm.nih.gov/, accessed on 15 October 2023) and the accession numbers were listed in Table 1.

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
