# Peer review of "Three New Species of Gongronella (Cunninghamellaceae, Mucorales) from Soil in Hainan, China Based on Morphology and Molecular Phylogeny"

_jof, 2023, doi:10.3390/jof9121182_

Round 1

Reviewer 1 Report

Comments and Suggestions for Authors

The study provides a valuable contribution to the understanding of the Gongronella genus, and the results are presented clearly and precisely. However, upon thorough review, I have identified some points where I suggest improvements to further enhance the work. All suggestions are distributed throughout the text. These recommendations aim to improve the clarity and impact of your work. I appreciate in advance your attention to these points and am available to clarify any doubts or discuss additional suggestions that may arise.

Author Response

Dear editors and referee,

Thank you for your valuable suggestion! In response to these questions, I am answering point-by-point as follows and highlighted in the revised version of our manuscript:

  1. Row 18: I suggest, for better writing of the article, replacing the phrase "more three new species" with "three more new species".

I accepted the suggestions.

  1. Row 22: I suggest, "classified" to "they were nominated as"...

Thanks. It is done.

  1. Row 24: I suggest modifying it to "presence of oil droplets in the sporangiospores" for greater clarity.

I accepted the suggestions.

  1. Row27: replace "to" with "from" to correct the preposition.

I accepted the suggestions.

  1. Row 34: Please insert italics.

I accepted the suggestions.

  1. Row38-39: I suggest making a small modification to the sentence: "Gongronella spp. were found to be among phosphate-solubilizing [5] and metalaxyl degrading fungi [6]," I would add a verb to improve fluidity, for example, "Gongronella spp. were found to possess phosphate-solubilizing abilities [5] and to degrade metalaxyl [6]."

I accepted the suggestions.

  1. Row45: please insert authors

I accepted the suggestions.

  1. Row 45: please insert authors

I accepted the suggestions.

  1. Row 46: please insert authors

I accepted the suggestions.

  1. Row 46: please insert authors

I accepted the suggestions.

  1. Row 49: Please insert italics.

Thanks. It is done.

  1. Row49: I suggest using "passed" instead of "past" to correct the grammar.

Thanks. It is done.

  1. Row 51: I would suggest adding "have been" before "the presence" to make the sentence more fluid.

I accepted the suggestions.

  1. Row 56: I suggest "include" to improve clarity.

Thanks. It is done.

  1. Row 57: would suggest adding a reference for treating the type species G. urceolifera as a synonym of G. butleri for greater transparency.

I have added the reference for this synonimization, that is “16. Peyronel, B. and G.D. Vesco. Ricerche sulla microflora di un terreno agrario presso Torino. Allionia, 1955. 2: 357–417.” Consequently, I have modified all relavant citation numbers in the manuscript.

  1. Row 77: I suggest specifying the exact amount of soil in order to make the methodology applied clear.  For example, "Exactly 10 g of soil samples were transferred into a conical bottle."

It is done.

  1. Row 79: I suggest "pipetted"

It is done.

  1. Row85: I suggest clarifying "d" for days.

I accepted the suggestions.

  1. Row91: I suggest replacing "with" with "in" makes the reading a little clearer.

I accepted the suggestions.

  1. Row 92-93:I suggest specifying that 30 measurements were taken for each morphological characteristic, for example, "Each morphological character was statistically calculated from 30 measurements per character [17]."

Thanks for your suggestion. It is done.

  1. Row 100-102: To improve the clarity of the sentence, I would suggest splitting it into two sentences to improve clarity, for example, "Genomic DNA was extracted from mycelia using the CTAB (cetyl trimethyl ammonium bromide) method. Additionally, a rapid extraction plant DNA magnetic beads kit was employed (GeneOn BioTech; http://www.geneonbio.com/content/?1153.html) [18, 19]."

Thanks for your suggestion. It is done.

  1. Row102: To improve the clarity of the sentence, I suggest adding "The internal transcribed spacer (ITS)".

Thanks for your suggestion. It is done.

  1. Row109: add "reaction mixture".

I accepted the suggestions.

  1. Row113:replacing “were observed with an ultraviolet of” with "were visualized under ultraviolet light at 254 nm".

Thanks for your suggestion. It is done.

  1. Row 123-125 - To improve understanding of the sentence, I suggest clarifying that the sequences were aligned using MAFFT 7 online services and providing details about the alignment, for example, "Newly generated sequences and their related sequences retrieved from GenBank (Table 1) were aligned using MAFFT 7 online services (http://mafft.cbrc.jp/alignment/server/, accessed on October 20, 2023) [24], employing default parameters for accurate alignment."

Thanks for your suggestion. It is done.

  1. Row143: replacing “as outgroup” with "selected as the outgroup for phylogenetic comparison".

It is done.

  1. Row 162: Please insert italics

It is done.

  1. In order to specify the characteristics and their dimensions clearly, I suggest separating the sporangiosopores characterized as ovoid and fusiform, as they present different morphologies without applying a single measurement.

Thank you very much! This suggestion led me to examine it carefully and found that sporangiospores of this strain were mostly oval and much less fusiform. Hence, we deleted the description for these fusiform ones due to its unrepresentativeness.

  1. Row 190-192:For a more precise description, I suggest providing more information, such as other temperatures and cultivation media, whether or not the isolates were kept under lighting during the colony growth period, for example... this is information that helps in replicating the methodology used for the identification of species of the proposed species.

Thanks for your suggestion. By reviewing the description of the closely related species, I found that the strains of Gongronella were most commonly cultured with PDA medium at 25℃ for 7 days. MEA medium at 25-27 °C was used in a small number of cases. Therefore, the culture conditions in this paper were set to PDA medium, dark environment, 25℃, and 7 days. Consequently, I added the dark culture information.

  1. Row206-207: replacing “The G. multiramosa is smaller in sporangium than G. pedratalhadensis” with "The sporangium of G. multiramosa is smaller than that of G. pedratalhadensis"

Thanks for your suggestion. It is done.

  1. Row207-208: replacing “ multiramosa has fewer septa on sporangiophores than G. pedratalhadensis ” with "G. multiramosa has fewer septa on sporangiophores compared to G. pedratalhadensis"

Thanks for your suggestion. It is done.

  1. Row 217: Insert italics

It is done.

  1. Row 224: I suggest a modification to the epithet, it is not clear what characteristic designates the proposed species (at least from the images provided).  Perhaps the place of isolation or even the substrate from which it was isolated provides a more appropriate epithet.

Thanks for your suggestion. I have renamed this species according to the sample collecting place.

  1. Row 235-241: replacing “and” with “,”

Thanks for your suggestion. It is done.

  1. Row 244-245: For a more precise description, I suggest providing more information, such as other temperatures and cultivation media, whether or not the isolates were kept under lighting during the colony growth period, for example... this is information that helps in replicating the methodology used. for the identification of species of the proposed species.

Thanks for your suggestion. By reviewing the description of the closely related species, I found that the strains of Gongronella were most commonly cultured with PDA medium at 25℃ for 7 days. MEA medium at 25-27 °C was used in a small number of cases. Therefore, the culture conditions in this paper were set to PDA medium, dark environment, 25℃, and 7 days. Consequently, I added the dark culture information.

  1. Row255-257: To make the reading more fluid and clear, I suggest modifying the excerpt to: "Due to the unavailability of descriptions and illustrations for G. banzhaoae, we compare only G. butleri and G. hydei with the new species G. parvicollariata."

Thanks for your suggestion. It is done.

  1. Row259: Please check the sentence. There may be an error in the values, as it is unusual to measure the size of colonies in millimeters. Generally, colony measurements are expressed in units such as square centimeters or millimeters. Did you want to express their height?

Thanks for your suggestion. Here I have adopted a vague description. The two strains differ considerably in colonial diameter and are therefore measured in millimeters. Now I have added on the article.

  1. Row299: Please see my suggestions for this feature in the description of Gongronella multiramosa (Previous).

Thanks for your suggestion. I've made a change.

  1. Row307-305: For a more precise description, I suggest providing more information, such as other temperatures and cultivation media, whether or not the isolates were kept under lighting during the colony growth period, for example... this is information that helps in replicating the methodology used. for the identification of species of the proposed species.

Thanks for your suggestion. By reviewing the description of the closely related species, I found that the strains of Gongronella were most commonly cultured in PDA medium at 25℃ for 7 days. MEA medium was used in a small number of cases, which were cultured at 25-27 °C. Therefore, the culture conditions described in this paper were selected as PDA medium, dark environment, 25℃, and 7 days.

  1. Row318: I suggest standardizing by quoting "days".

Thanks for your suggestion. I've made a change.

  1. Row323: Please see my previous suggestion (standardize citing "days").

Thanks for your suggestion. I've made a change.

  1. Row339-340: Please check the sentence, perhaps the expression "booming period" could be informal. It may be preferable to use more objective terms, such as "an increase in research activity" or "a period of significant growth."

Thanks for your suggestion. It is done.

  1. Row340: The word "hitherto" can be replaced with "currently" for clarity.

Thanks for your suggestion. It is done.

  1. Row349: For greater clarity I suggest rephrasing the sentence to: for greater clarity: "constructed in this study without the inclusion of G. pamphilae sequences."

Thanks for your suggestion. It is done.

  1. Row353-354: For greater clarity I suggest rephrasing the sentence to: "Since then, taxonomists studying this genus have adopted the practice of combining ITS-LSU rDNA with morphology".

It is done.

  1. Row359-360: replacing “while pedratalhadensis is quite the opposite, with zygospores and no chlamydospores.” with "In contrast, G. pedratalhadensis has zygospores but lacks chlamydospores."

Thanks for your suggestion. It is done.

  1. Row363-364: For greater clarity, I suggest rephrasing the sentence to: "Lastly, G. parvicollariata lacks obvious collars, while G. butleri and G. hydei exhibit remarkable collars."

Thanks for your suggestion. It is done.

Sincerely yours,

Yi-Xin Wang

Reviewer 2 Report

Comments and Suggestions for Authors

In this study, the authors describe three new Gongronella species isolated from soil in China. Every description of undiscovered biodiversity is a valuable contribution to general knowledge of mycobiota. Although the text is general well structured, I’d recommend some modifications. My main concern is about paraphyly of type species G. butleri. In the presented form type strain of G. butleri is closer related to G. parvicollariata than to another G. butleri strain. I would acknowledge some wider discussion of this issue. For example, was the barcoding gap for species delimitation calculated?

The introduction could be improved by being more focused on species diversity that on biotechnological applications of Gongronella. The Authors don’t focus on potential application of their strains, so general diversity introduction would be sufficient in this case.

Instead I’m interested whether Gongronella sequences could be detected in eDNA databases like theglobalfungi. Maybe this approach could increase our knowledge on diversity of poorly studied and rarely isolated group of fungi.

I attach file with some more detailed comments.

Author Response

Dear editors and referee,

Thank you for your valuable suggestion. In response to these questions, I answer as follows and highlighted in the revised version:

  1. Row 18: the order of “more three new species”

Response: Thanks for your suggestion. I changed the order to "with "three more new species".

  1. Row 19: phylogenetic information?

Response: Thanks for your suggestion. I changed the “gene marks” to “phylogenetic information”.

  1. Row 22: replacing “nominated as” with “named”?

Response: Thanks for your suggestion. I replaced “nominated as” with “classified as”.

  1. Row 41: In other cases full scientific name with authorship was provided

Response: Thank you for your valuable suggestion. I have added this information.

  1. Row 48-50: not needed “1955”, “1962”.

Response: Thank you for your valuable suggestion and I deleted them.

  1. Row 72: Fungal Name numbers, etymologies, typifications, descriptions and illustrations for these three new species

Response: Thank you for your valuable suggestion. It is done.

  1. Row 74: morphological observations? measurements?

Response: Thank you for your valuable suggestion. I have revised the “morphology” as “morphological observations” following your advice.

  1. Row 122: the database was searched for sth using blast algorithm

Response: Thank you for your valuable suggestion. I have revised this sentence as “The obtained DNA sequences were searched using BLAST with default parameters for closely related items against NCBI GenBank nucleotide database”.

  1. Row 154: Gongronella butleri is paraphyletic on this tree and type strain is closer to your specimens than to other G. butleri. Could you explain this?

Response: Thank you for your valuable suggestion. In fact, the two ITS sequences of G. botleri differ by only two bases, whereas the new species described here differs by 12 bases from the G. botleri type species. Moreover, the ML phlogram was selected for presentation, while in BI phylogram, G. bartleri clustered together and was separated from G. quarhanesis (G. parvicollariata).

  1. Row 162 - italics

Response: Thank you for your valuable suggestion. It is done.

  1. Row 214: this is rather not character relevant for identification as formation of zygospores probably depend on other factors and hopefully under some conditions both species would form zygospores

Response: Thank you for your valuable suggestion. I deleted the comparison about zygospores.

  1. Row 217: italics

Response: Thank you for your valuable suggestion. It is done.

  1. Row 338: a little bit exaggerated... what about Rozelids or water fungi? Or Trichomerium like?

Response: Thank you for your valuable suggestion. I replaced “the most understudied taxa” with “one of the most understudied taxa”.

  1. Row 347: Have you tried to fish similar sequences from global soils amplicons datasets? like from theglobalfungi database? This could improve our knowledge of distribution.

Response: Thank you for your valuable suggestion. I have searched in the GlobalFungi database and retrieved 3341 records about the genus Gongronella and they are distributed worldwide too. So, I have put the information in the revised manuscript as “which was confirmed by 3341 samples and 25840 sequence variants filtered from the GlobalFungi database (https://globalfungi.com/, accessed on Dec 6, 2023; Asia 72.13%, North America 12.57%, South America 6.55%, Europe 3.32%, Africa 2.87%, Austrlia 2.39%, and Pacific Ocean 0.15%)”.

  1. Row356: What abaout G. gutleri paraphyly on your tree?

Response: Thank you for your valuable suggestion. In fact, the two ITS sequences of G. botleri differ by only two bases, whereas the new species described here differs by 12 bases from the G. botleri type species. Moreover, the ML tree was selected for presentation, while in BI tree, G. bartleri clustered together and was separated from G. quarhanesis (G. parvicollariata).

  1. Row 359: I don't recommend to treat presence or absence of sexual reproduction as taxonomically important character.

Response: Thank you for your valuable suggestion. I have deleted these comparisons.

  1. Row 371: This is far edged and not needed in taxonomic paper. You don't provide any biotechnological tests for this.

Response: Thank you for your valuable suggestion. I have deleted relevant sentences.

Sincerely yours,

Yi-Xin Wang
